# Cotton Yield Estimation Using the Remotely Sensed Cotton Boll Index from UAV Images



Guanwei Shi [1,2], Xin Du [1], Mingwei Du [3], Qiangzi Li [1,*], Xiaoli Tian [3,*], Yiting Ren [1,2], Yuan Zhang [1] and Hongyan Wang [1]

1 Aerospace Information Research Institute, Chinese Academy of Sciences, Beijing 100094, China
2 University of Chinese Academy of Sciences, Beijing 100049, China
3 State Key Laboratory of Plant Physiology and Biochemistry, Key Laboratory of Crop Cultivation and Farming System, Center of Crop Chemical Control, China Agricultural University, Beijing 100193, China
* Correspondence: liqz@radi.ac.cn (Q.L.); tianxl@cau.edu.cn (X.T.); Tel.: +86-010-6485-5094 (Q.L.); +86-010-6273-4550 (X.T.)

**Abstract:** Cotton constitutes 81% of the world's natural fibers. Accurate and rapid cotton yield estimation is important for cotton trade and agricultural policy development. Therefore, we developed a remote sensing index that can intuitively represent cotton boll characteristics and support cotton yield estimation by extracting cotton boll pixels. In our study, the Density of open Cotton boll Pixels (DCPs) was extracted by designing different cotton boll indices combined with the threshold segmentation method. The relationship between DCP and field survey datasets, the Density of Total Cotton bolls (DTC), and yield were compared and analyzed. Five common yield estimation models, Linear Regression (LR), Support Vector Regression (SVR), Classification and Regression Trees (CART), Random Forest (RF), and K-Nearest Neighbors (KNN), were implemented and evaluated. The results showed that DCP had a strong correlation with yield, with a Pearson correlation coefficient of 0.84. The RF method exhibited the best yield estimation performance, with average $R^2$ and rRMSE values of 0.77 and 7.5%, respectively (five-fold cross-validation). This study showed that RedGreenBlue (RGB) and Near Infrared Red (NIR) normalized, a normalized form index consisting of the RGB and NIR bands, performed best.

**Keywords:** cotton; UAV; cotton boll index; threshold segmentation; yield estimation

## 1. Introduction

Cotton is the main fiber crop worldwide. According to statistics released by the Food and Agriculture Organization (FAO) of the United Nations, cotton accounted for 26 million tons of fiber production in the period from 2018 to 2019 (specifically, 1 August 2018 to 31 July 2019) [1]. It represented 81 percent of natural fiber production and 24 percent of total fiber production. Accurate and rapid estimation of cotton yield is of great significance to cotton trade and agricultural policy planning, particularly in the context of population growth and climate change.

Traditional cotton yield estimation methods require manual sampling, which is very labor intensive and time-consuming. These methods are increasingly unsuitable for practical applications. Since remote sensing technology can quickly obtain a wide range of information from ground objects, it is widely used for cotton yield estimation. Common remote sensing platforms include satellites and Unmanned Aerial Vehicles (UAVs).

Due to rich data sources and optimal data processing procedures, satellite remote sensing was first applied to cotton yield estimation. Liu et al. [2] used the time series Enhanced Vegetation Index (EVI) obtained from the HJ-1A/B satellite to rank cotton yields in a county. Alganci et al. [3] used the time series satellite images obtained by Système Pour l'Observation de la Terre 5 (SPOT5), Thematic Mapper (TM), and Enhanced Thematic Mapper (ETM) to calculate the vegetation index and combined them with measured digital

photos and meteorological data to estimate the cotton yield according to the empirical formula. Leon et al. [4] used the measured yield data of cotton in sampling grids and the vegetation index calculated by satellites to carry out regression modeling, and the correlation coefficient in the middle and late stages of crop growth reached 0.87. Based on the Normalized Difference Vegetation Index (NDVI), a quadratic regression model was applied to remotely-sensed data and the estimated cotton yield [5]. Prasad et al. [6] used a time-series 16-day composite of Moderate Resolution Imaging Spectroradiometer (MODIS) NDVI data to extract seasonality parameters. A multiple regression model constructed with seasonality parameters was applied to estimate cotton yield. Although satellite data have a wide application, they suffer from low spatial resolution. This makes satellite remote sensing unsuitable for accurate yield estimation on the farm scale. In addition, the less frequent revisit of satellite remote sensing limits data acquisition on time phases important for yield estimation.

UAV remote sensing has the advantages of flexibility, high spatial resolution, and customizable data acquisition time [7]. These features overcome some of the limitations of satellite remote sensing, leading to the rapidly increased use of UAV images in cotton yield estimation.

Many factors associated with the physiological and biochemical state of cotton plants have been used in yield estimation based on UAV images, mainly including vegetation indices, canopy cover information, and Plant Height (PH). Huang et al. [8] implemented modeling with the Ratio Vegetation Index (RVI) and cotton yield, and the $R^2$ value reached 0.47. Yeom et al. used vegetation indices obtained from time series images in a cotton crop growth model to estimate yield [9]. Feng et al. constructed eight indicators using two-phase UAV images and then used a multiple regression model based on these indicators to estimate cotton yield [10]. Feng et al. also evaluated the performance of a UAV-based remote sensing system with a low-cost RGB camera to estimate cotton yield based on PH [11]. Chu [12] used cotton PH and canopy cover information to calculate cotton yield according to an empirical formula, which was retrieved from point cloud-based digital surface models (DSMs) and orthomosaic images. Ma et al. [13] used vegetation indices and texture features extracted from the ultra-high-resolution RGB images obtained by UAVs to estimate cotton yield.

The above methods mainly focused on the state of the cotton plant, but the cotton yield is primarily produced by cotton bolls, which are organs of cotton and not the whole plant. Yield estimation methods based on cotton boll extraction are more straightforward than focusing on the whole cotton plant. Since the degree of mechanization of cotton production is increasing and defoliation and ripening have become necessary conditions for cotton production [14], cotton bolls are obvious enough in UAV images to be extracted. Based on this, many studies have focused on the extraction of cotton bolls. Laplace image transformation was used to establish the boll coverage of plots to estimate yield, and the $R^2$ reached 0.83 after removing outliers [15]. Threshold segmentation and morphological filtering have also been used to extract the cotton boll area in UAV images [16]. Yeom used this method to build a linear regression model for cotton boll area and yield in different irrigation plots, and the $R^2$ was between 0.63 and 0.65. Fue et al. [17] used the RGB color threshold to separate each RGB component of the image. The white components of the image can be masked as cotton bolls. Wei et al. [18] used threshold segmentation in HSV color space to obtain cotton bolls. Classification algorithms were also used to identify cotton bolls [19,20]. Sun et al. [21] used a robot platform equipped with a digital camera to detect and count the number of bolls based on an operation that splits a single boll into two or more disjoint regions. Zhang collected near-infrared UAV images of cotton fields. An object-oriented method was used to segment cotton bolls and backgrounds. Xu et al. [22] established a cotton yield estimation model based on time series UAV remote sensing indices combined with boll opening pixel percentages extracted by U-Net. The time series data of various vegetation indices and the proportion of open cotton bolls were input into the Back Propagation (BP) neural network as training data, and the $R^2$ reached 0.853.

Previous studies mainly adopted spectral reflectance and common indices that are sensible of vegetation rather than cotton bolls, seldom considering the distinctive features of bolls in yield estimation. That resulted in many uncertainties when cotton bolls' spectral features were not as strong as leaves. Meanwhile, there are too many spectral bands and indices involved, which leads to compute-intensive tasks.

In this study, a more reasonable method was proposed to achieve definitive yield estimation, which aims to develop a novel spectral index, emphasizing the spectral characteristics of cotton bolls, to improve yield modeling accuracies. The index considered the spectral differences between bolls and other field targets deeply and could support cotton yield estimation using a naive model or as an input contributing deep learning models together with raw bands as input.

The main tasks of the study mainly involved three parts. The first one was the development of the index that can intuitively represent the cotton boll characteristics. Another one was cotton boll information extraction using the threshold segmentation method referring to ground survey data. Additionally, the last one was an experiment on cotton yield estimation considering the cotton boll information from the index. The study area and data are described in Section 2. The method used in this paper is introduced in Section 3. The results and performance assessment are provided in Section 4. Section 5 is the discussion, and Section 6 is the conclusion.

## 2. Study Area and Data

This section presents the study area and data.

### 2.1. Study Area

The study area is located in Cangzhou, Hebei Province, which is one of the main cotton-producing areas in China (Figure 1).

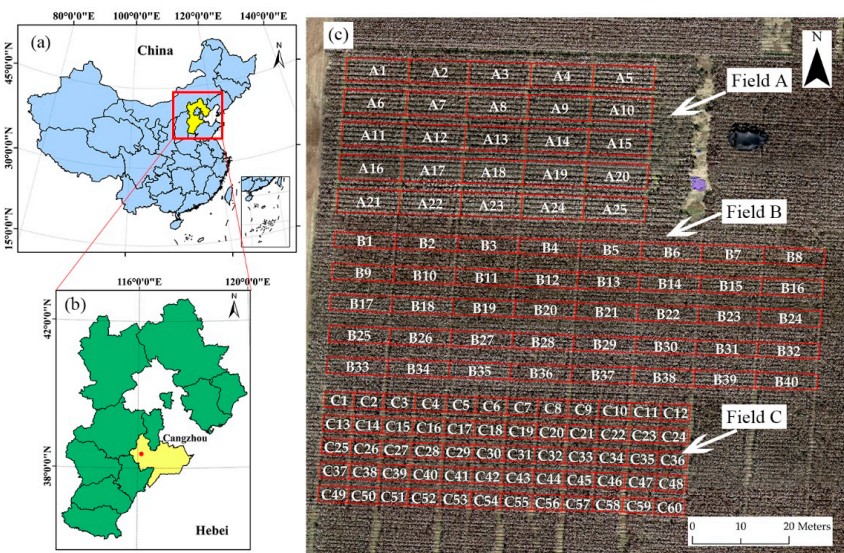

**Figure 1.** (**a**) Location of Hebei Province in China; (**b**) location of Cangzhou city and the experimental area in Hebei Province; (**c**) overview of the 125 experimental plots in the cotton field. Each plot has an independent code.

A total of 125 experimental plots of three different sizes were set up in the experimental field. The theoretical planting areas were 68.00 m$^2$ (Filed A), 54.72 m$^2$ (Filed B), and 18.24 m$^2$ (Filed C). Three fields with different areas are shown in Figure 1. Guoxin 26, a representative cotton variety, was planted in the study area, provided by the Hebei Cotton Seed Engineering Research Center.

Cangzhou's climate conditions are suitable for cotton cultivation. Cangzhou has four different seasons and a continental semiarid monsoon climate, with an average annual

temperature of 12.2 °C, a minimum temperature of −25 °C, and a maximum temperature of 43 °C. The average annual rainfall is 549.5 mm. The rainfall is unevenly distributed throughout the year, and 80% falls in the summer [23].

*2.2. Data*

The data used in this study are divided into UAV data and field survey data.

### 2.2.1. UAV Data

A DJI M300 with an integrated ALTUM sensor made by Micasense was used to collect the UAV data. The DJI M300 RTK integrates an RTK module that has a strong anti-magnetic interference capability and precise positioning capability. The camera collected 1280 × 960 pixels per image band. The specific spectral bands were blue at approximately 475 nm, green at 560 nm, red at 668 nm, Red-Edge (RE) at 717 nm, Near Infrared Red (NIR) at 840 nm, and thermal at 8–14 μm.

Data were acquired between 11:00 and 13:00 local time on 22 October 2021. A flight altitude of 50 m was chosen, and a Ground Sampling Distance (GSD) of 2.16 cm was achieved. The heading overlap rate was 80%, and the side overlap rate was 75%. Radiometric correction and alignment of the data were performed in Agisoft Metashape 1.7.0 (https://www.agisoft.com). The sun sensor and the reflectance panel were used for radiometric correction. We obtained blue/green/red/RE/NIR reflectance images and thermal-infrared emissivity images of the experimental area. The sensor parameters are shown in Table 1.

**Table 1.** Sensor parameters.

| Parameter | Multispectral Bands | Thermal Infrared Band |
|:---:|:---:|:---:|
| Resolution | 2064 × 1544 | 160 × 120 |
| Field of View (FOV) | 48° × 36.8° | 57° × 44.3° |
| Flight altitude | 50 m | 50 m |
| Ground sampling (GSD) | 2.16 cm/px | 13.56 cm/px |
| Imaging band | B (475 nm), G (560 nm), R (668 nm), RE(717 nm), NIR (840 nm) | 8–14 μm |

The data acquired by the UAV were displayed as an RGB image, as shown below (Figure 2).

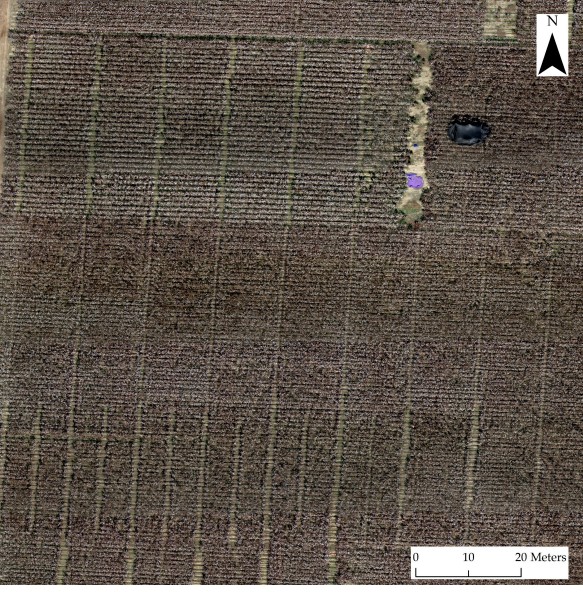

**Figure 2.** Original image of the study area.

### 2.2.2. Field Survey Data

Through random sampling and manual counting, the Number of Total open Cotton bolls (NTC) from 10 (18.24 m$^2$, 54.72 m$^2$) or 15 (63.00 m$^2$) cotton plants were obtained. Because the total number of cotton plants in the experimental plot was known, the NTC was calculated.

Cotton in all plots was harvested by hand, and the production corresponding to each experimental plot was obtained through actual measurement. The field survey data are shown in Appendix A, which is divided into three tables corresponding to three different size plots (Tables A1–A3).

All the ground observations and UAV data were acquired on 22 October 2021 (Figure 3).

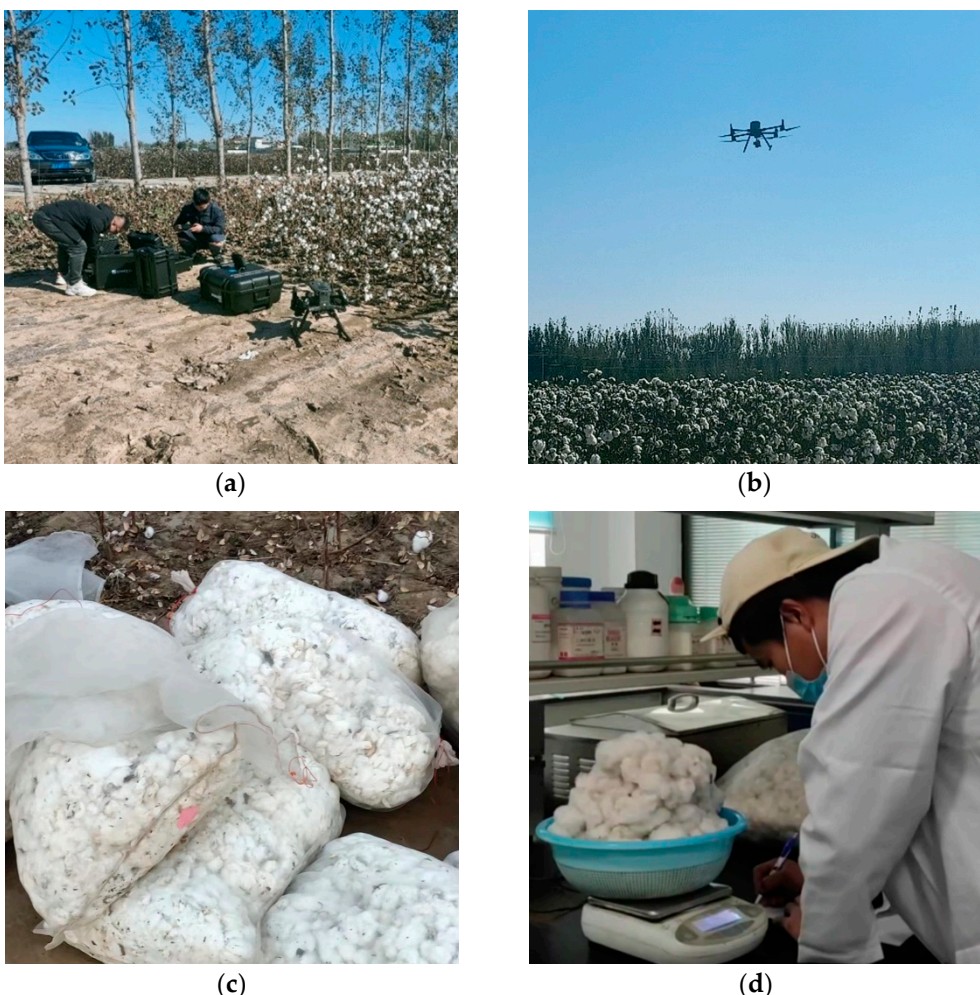

**Figure 3.** Acquisition of UAV data and field survey data. (**a**) Flight preparation; (**b**) data collection; (**c**) partition packaging; (**d**) weighing production.

## 3. Methods

The reflectivity bands (blue, green, red, red-edge, and near-infrared) from the orthomosaic image were used to calculate cotton boll indices that highlighted the spectral characteristics of the cotton boll. Adaptive threshold segmentation was applied to extract cotton boll pixels. The extracted results and field survey data were assessed using correlation analysis, and the model was built according to the results of the correlation analysis. The method was divided into three parts: (1) cotton boll index calculation, (2) extraction of cotton bolls and correlation analysis, and (3) modeling and performance assessment (Figure 4).

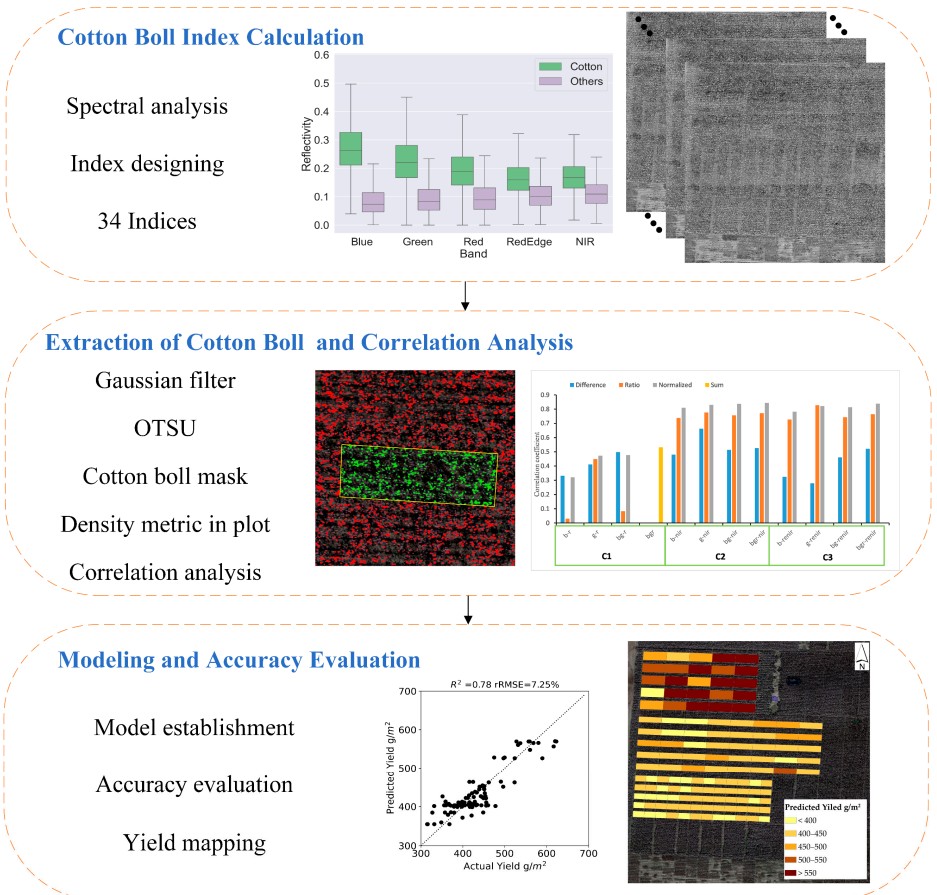

**Figure 4.** Overall methodology workflow.

### 3.1. Cotton Boll Index Calculation

Spectral analysis using UAV data from the cotton pre-harvest period was performed. The spectral differences between bolls and other ground objects at the time of harvest were analyzed. Then, we designed the boll index based on the results of the spectral analysis.

#### 3.1.1. Spectral Analysis

The visual characteristics of the cotton bolls were analyzed. Cotton bolls are visually white, indicating that they have high reflectance in the visible light band (blue, green, and red). If an index is designed to highlight the high reflectivity feature in the visible light band, it will have a better performance for extracting bolls. This was confirmed by the results of the quantitative analysis of the boxplot (Figure 5). Boxplots are widely used for visualizing the distribution of continuous unimodal data [24,25].

The reflectivity of cotton bolls in the five bands was different from that of non-cotton-boll objects. From short to long wavelengths, there was a consistent downward trend in cotton boll pixels, and the reflectivity in the near-infrared band was slightly higher. However, non-cotton-boll objects exhibited an upward trend from shortwave to longwave radiation.

Different reflectivity characteristics can be used to design a cotton boll index that distinguishes open cotton bolls from the background. Cotton bolls have high reflectivity in visible light and low reflectivity in the infrared band. On the contrary, non-cotton boll targets have high reflectivity in the infrared band and low reflectivity in the visible light band. The reflectance data obtained by the boxplot analysis are shown in Table 2:

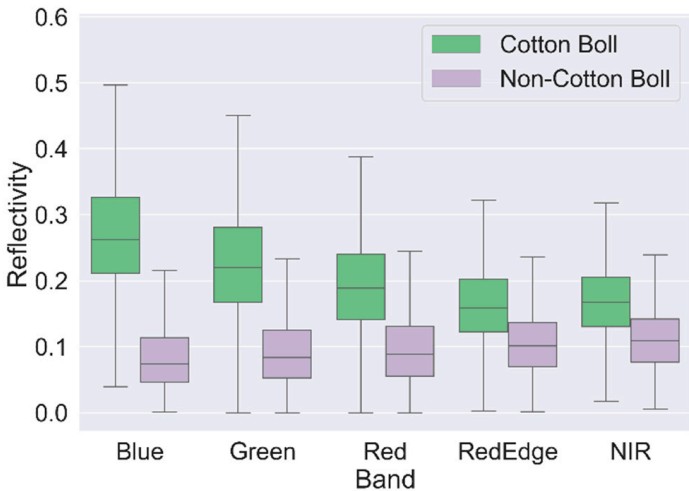

**Figure 5.** Distribution of reflectance for 65,603 cotton boll pixels and 118,996 non-cotton boll pixels extracted by visual interpretation from the multispectral image.

**Table 2.** The reflectance data obtained by the boxplot analysis. C for cotton and O for others.

| | Blue | | Green | | Red | | Red-Edge | | NIR | |
|---|---|---|---|---|---|---|---|---|---|---|
| | C | O | C | O | C | O | C | O | C | O |
| Mean | 0.27 | 0.08 | 0.23 | 0.09 | 0.19 | 0.10 | 0.16 | 0.10 | 0.17 | 0.11 |
| STD | 0.08 | 0.05 | 0.08 | 0.05 | 0.07 | 0.05 | 0.06 | 0.05 | 0.05 | 0.05 |
| 25% | 0.21 | 0.05 | 0.17 | 0.05 | 0.14 | 0.06 | 0.12 | 0.07 | 0.13 | 0.08 |
| 50% | 0.26 | 0.07 | 0.22 | 0.08 | 0.19 | 0.09 | 0.16 | 0.10 | 0.17 | 0.11 |
| 75% | 0.33 | 0.11 | 0.28 | 0.13 | 0.24 | 0.13 | 0.20 | 0.14 | 0.21 | 0.14 |

### 3.1.2. Index Design

Remote sensing indices are algebraic combinations of several spectral bands designed to highlight objects [26]. The most widespread type of index used is the mathematical combination of reflectance bands. Many scientists have proposed different forms of indices [27–29], including normalized, ratio, difference, and summation. This study also adopts these methods for constructing a remote sensing index.

It can be seen from the spectral analysis that the reflectance of the cotton bolls and other targets is significantly different. Cotton bolls have high reflectivity in the visible bands (red, green, and blue) and low reflectivity in the infrared bands (red-edge and near-infrared). On the contrary, other targets have high reflectivity in the infrared band and low reflectivity in the visible light band. To highlight this feature, three arithmetic methods for visible and infrared bands were difference (d), ratio (r), and normalization (n). Through the index calculation, the pixels of cotton bolls have high values, and the pixels of other objects have low values.

In order to explore whether the red-edge band has an effect on the index performance, the red-edge band did not participate in the construction of all indices.

In order to explore whether it is possible to highlight cotton bolls in the visible light band only, an attempt was made to construct some indices with the red band instead of the infrared information. This is because the reflectance performance of the red band in the spectral analysis is relatively closer to the infrared information than the blue/green band.

In visible bands, the difference between cotton bolls and other targets in the blue and green bands is larger than that in the red band. Some indices were designed to explore whether the goal can be achieved without using the red band.

Thirty-four indices were designed. Considering the band acquisition capabilities of different sensors, three combinations were divided. Four methods for combining forms across bands were established. The three combinations were as follows: only visible light

bands (Combination 1, C1), the combination of visible light bands and near-infrared bands (Combination 2, C2), and the combination including visible light bands, near-infrared bands, and red-edge bands (Combination 3, C3). The four methods for combining bands were the difference (d), ratio (r), normalization (n), and summation (sum).

In Table 3, the calculation formulas and abbreviations for all indices are listed.

**Table 3.** All indices, including the three combinations (C1, C1, and C3.) The four methods for combining bands were the difference (d), ratio (r), normalization (n), and summation (sum).

| C1 | | C2 | | C3 | |
|---|---|---|---|---|---|
| $\text{Blue} - \text{Red}$ | b-r_d | $\text{Blue} - \text{NIR}$ | b-nir_d | $\text{Blue} - (\text{RE} + \text{NIR})$ | b-renir_d |
| $\frac{\text{Blue}}{\text{Red}}$ | b-r_r | $\frac{\text{Blue}}{\text{NIR}}$ | b-nir_r | $\frac{\text{Blue}}{\text{RE}+\text{NIR}}$ | b-renir_r |
| $\frac{\text{Blue}-\text{Red}}{\text{Blue}+\text{Red}}$ | b-r_n | $\frac{\text{Blue}-\text{NIR}}{\text{Blue}+\text{NIR}}$ | b-nir_n | $\frac{\text{Blue}-(\text{RE}+\text{NIR})}{\text{Blue}+\text{RE}+\text{NIR}}$ | b-renir_n |
| $\text{Green} - \text{Red}$ | g-r_d | $\text{Green} - \text{NIR}$ | g-nir_d | $\text{Green} - (\text{RE} + \text{NIR})$ | g-renir_d |
| $\frac{\text{Green}}{\text{Red}}$ | g-r_r | $\frac{\text{Green}}{\text{NIR}}$ | g-nir_r | $\frac{\text{Green}}{\text{RE}+\text{NIR}}$ | g-renir_r |
| $\frac{\text{Green}-\text{Red}}{\text{Green}+\text{Red}}$ | g-r_n | $\frac{\text{Green}-\text{NIR}}{\text{Green}+\text{NIR}}$ | g-nir_n | $\frac{\text{Green}-(\text{RE}+\text{NIR})}{\text{Green}+\text{RE}+\text{NIR}}$ | g-renir_n |
| $\text{Blue} + \text{Green} - \text{Red}$ | bg-r_d | $\text{Blue} + \text{Green} - \text{NIR}$ | bg-nir_d | $\text{Blue} + \text{Green} - (\text{RE} + \text{NIR})$ | bg-renir_d |
| $\frac{\text{Blue}+\text{Green}}{\text{Red}}$ | bg-r_r | $\frac{\text{Blue}+\text{Green}}{\text{NIR}}$ | bg-nir_r | $\frac{\text{Blue}+\text{Green}}{\text{RE}+\text{NIR}}$ | bg-renir_r |
| $\frac{\text{Blue}+\text{Green}-\text{Red}}{\text{Blue}+\text{Green}+\text{Red}}$ | bg-r_n | $\frac{\text{Blue}+\text{Green}-\text{NIR}}{\text{Blue}+\text{Green}+\text{NIR}}$ | bg-nir_n | $\frac{\text{Blue}+\text{Green}-(\text{RE}+\text{NIR})}{\text{Blue}+\text{Green}+\text{RE}+\text{NIR}}$ | bg-renir_n |
| $\text{Blue} + \text{Green} + \text{Red}$ | bgr_sum | $\text{Blue} + \text{Green} + \text{Red} - \text{NIR}$ | bgr-nir_d | $\text{Blue} + \text{Green} + \text{Red} - (\text{RE} + \text{NIR})$ | bgr-renir_d |
| | | $\frac{\text{Blue}+\text{Green}+\text{Red}}{\text{NIR}}$ | bgr-nir_r | $\frac{\text{Blue}+\text{Green}+\text{Red}}{\text{RE}+\text{NIR}}$ | bgr-renir_r |
| | | $\frac{\text{Blue}+\text{Green}+\text{Red}-\text{NIR}}{\text{Blue}+\text{Green}+\text{Red}+\text{NIR}}$ | bgr-nir_n | $\frac{\text{Blue}+\text{Green}+\text{Red}-(\text{RE}+\text{NIR})}{\text{Blue}+\text{Green}+\text{Red}+\text{RE}+\text{NIR}}$ | bgr-renir_n |

### 3.2. Extraction of Cotton Boll and Correlation Analysis

Band math was applied to UAV orthomosaic images to obtain the cotton boll index introduced in Section 3.1, and the spectral reflection characteristics of the open cotton boll were highlighted on the index map. A gaussian filter was used on the cotton boll index to remove speckle noise. Gaussian filters are linear smoothing filters often used to reduce noise during image processing. This process uses a pixel-level weighted average approach. The value of each pixel is calculated as a weighted average of the pixel and the adjacent pixels, which can effectively suppress normal noise. A Gaussian filter was applied to the index image to produce a smooth image to combat the distributed noise present in the cotton boll index grayscale map. The equation for the 2D Gaussian filter is [30]:

$$G(x,\ y) = \frac{1}{2\pi\sigma^2}e^{-\frac{x^2+y^2}{2\sigma^2}} \tag{1}$$

where x and y represent the distance between the central pixel and its neighbors, and σ represents the standard deviation. In the experiment, we chose the convolution kernel of (3,3) with a standard deviation of 0.8.

The Otsu method was then applied for global threshold segmentation. The Otsu algorithm was proposed by the Japanese scholar Otsu in 1979 and is known as the Otsu method, as well as the maximum interclass variance method [31]. The basic principle is to divide the image into two parts, foreground and background, according to the grayscale characteristics of the image. For the best threshold, the difference between the two parts should be the largest. We used the maximum inter-class variance as the criterion in the Otsu algorithm to assess the difference. The result of the segmentation was binarized to obtain an open cotton boll mask.

We used correlation analysis to quantitatively evaluate cotton boll masks generated from 34 boll indices. Correlation is often used to determine if there is an association between two observed variables and to estimate the strength of this relationship. The Pearson correlation is a measure of the linear association between two normally distributed random variables and is commonly denoted as r [32]. It is defined as follows:

$$r = \frac{\sum (X - \overline{X})(Y - \overline{Y})}{\sqrt{\sum (X - \overline{X})^2 \sum (Y - \overline{Y})^2}}, \tag{2}$$

where X and Y are two random variables, and $\overline{X}$ and $\overline{Y}$ are their mean values.

The density metric was constructed before the correlation analysis was implemented. Because the area of the experimental plots was different, the Number of open Cotton boll Pixels (NCP) in each plot was converted into a metric that represents NCP per square meter, which aims to evaluate the average cotton boll density in different plots. The NCP in different plots was divided by the area of the experimental plots to obtain the DCP.

$$\text{DCP} = \frac{\text{NCP}}{\text{the area of the plot}} \left(1/\text{m}^2\right). \tag{3}$$

DCP is involved in correlation analysis with field survey data, DTC, and yield. The better the correlation between the DCP and the field survey data, the better the index used to produce the boll mask is at highlighting the characteristics of the cotton boll. The statistical process of DCP is shown in Figure 6.

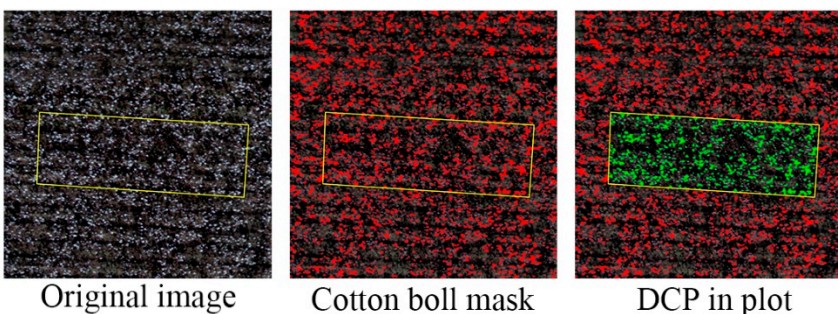

**Figure 6.** The yellow frame is the boundary of the experimental plot, the red mask is generated by the open cotton boll index, and the green pixels are the pixels that DCP needs to count. The boundary of the experimental plots was applied to the zonal statistics of the cotton boll mask to obtain DCP in each plot.

### 3.3. Modeling and Accuracy Evaluation

Since the goal of this paper was to use only the cotton boll index to estimate cotton yield, yield data from a field survey were used as the output data for the model. DCP represents the average flocculation condition of the experimental plot. DCP indicates how many boll pixels there are in a plot. The larger the value is, the better the plot is for harvesting. DCP was used as the input data of the model, which is an indicator generated by the index.

Cross-validation (CV) was used to evaluate the model and for hyperparameter selection. It can generate a reliable assessment of model performance and reduce contingency [33]. The idea of k-fold cross-validation is to leave out a certain number of samples at a time, perform the validation, and repeat the process k times. Our data were all manually sampled, so the amount of data was small (125 sets). The value of k was set to 5 so that 25 sets of data were left out at a time for training.

We implemented five algorithms commonly used in yield estimation: Linear Regression (LR), Support Vector Regression (SVR), Classification and Regression Trees (CART), Random Forest (RF), and K-Nearest Neighbors (KNN). All model hyperparameters were determined through the grid search via cross-validation.

The optimal hyperparameters of the models are shown in Table 4.

**Table 4.** Optimal hyperparameters after the grid search.

| Model | Optimal Hyperparameters after Research Grid |
|---|---|
| Support Vector Machines | Kernel: linear, C: 22 |
| Random Forest | max_depth: 3, max_features: auto, min_samples_leaf: 4, n_estimators: 60 |
| CART | max_depth: 4, min_samples_leaf: 4, splitter: best |
| KNN | n_neighbors: 14 |

### 3.3.1. Linear Regression

Many scholars use LR to establish the yield estimation model, and the model exhibits good estimation performance [34–36].

The equation for simple linear regression is of the form:

$$Y = a_0 + a_1 X + \varepsilon, \tag{4}$$

where Y is the response variable, X is the predictor variable, $a_0$ and $a_1$ are the regression coefficients or regression parameters, and $\varepsilon$ is an error term to account for the discrepancy between the predicted data from Equation (4) and the observed data.

For a dataset $(x_i, y_i)$ in which i = 1, 2, ... , n, the optimal parameters $a_0$ and $a_1$ are usually estimated by the least squares method, and its goal is to minimize the sum of the squares of $\varepsilon$:

$$\sum_{i=1}^{n} \varepsilon_i^2 = \sum_{z=1}^{n} (y_i - \alpha_0 - a_1 x_i)^2. \tag{5}$$

### 3.3.2. Support Vector Regression

SVR is an extension of Support Vector Classification (SVC) proposed by Boser [37]. SVR is an algorithm for estimating the relationship between the system input and output based on the available samples or training data. Input data and output data are jointly represented by an n-dimensional space. The objective of the support vector machine algorithm is to identify a hyperplane in the space that distinctly represents the relationship between input data and output data.

The vital step in SVR is the selection of the kernel function, which is often used to translate the dataset from a low- to high-dimensional space to better express the relationship between input data and output data. Various kernel functions were previously proposed and used in a wide variety of applications, such as the linear, polynomial, radius basis, and sigmoid functions.

The linear kernel was used in our regression experiments to construct a linear regression model. SVR and the simple linear regression described in Section 3.3.1 represent the predictive performance of the linear model.

### 3.3.3. Classification and Regression Trees

CART are machine-learning methods for constructing estimation models from data [38]. CART use a nonparametric regression technique that develops a decision tree based on a binary partitioning algorithm, which divides the current sample data into a left subtree and a right subtree until every leaf node is homogeneous or the function used to measure the quality of a split is minimized.

The CART method is widely used in the field of parametric regression [39,40]. Therefore, it has been integrated into many machine learning libraries. We used the Decision-TreeRegressor in the sklearn library 0.24 (https://scikit-learn.org/stable/, accessed on 1 September 2022).

### 3.3.4. Random Forest

RF is another popular ensemble learning method for both regression and classification problems [41]. Ensemble learning is an approach to machine learning. Multiple models are trained for the same problem, and the average of the multiple models is used to improve the predictive accuracy and control overfitting.

RF is a bagging ensemble learning method. It uses bootstrap technology to randomly sample from the original training sample data, generate a new training sample combination, and combine the decision trees generated by each sample set into a decision forest. The decision result is determined by the number of votes of all decision trees. The key parameter of the model is n, which is the number of trees in the forest, and n was set as 60 to distinguish it from CART.

### 3.3.5. K-Nearest Neighbors

KNN predicts the value of a target variable based on the similarity between the target value and its spatial neighbors [42]. It is a nonparametric model whose most critical hyperparameters are the number of neighbors (n) and the number of adjacent data points.

The n value was set to 14, and Euclidean distance was the criterion for distance judgment. The mean of the 14 nearest sample points was assigned to the regressor as a predicted value.

### 3.3.6. Accuracy Evaluation

In this paper, the performance of the models was assessed using two different metrics: the relative root mean square error (rRMSE) and determination coefficient ($R^2$).

$$\text{rRMSE} = \frac{\sqrt{\frac{1}{n}\sum_{i=1}^{n}(\hat{y}_i - y_i)^2}}{\overline{y}_i} \times 100, \tag{6}$$

$$R^2 = 1 - \frac{\sum(\hat{y}_i - y_i)^2}{\sum(y_i - \overline{y}_i)^2} \tag{7}$$

where $\hat{y}_i$ is the predicted value, $y_i$ is the actual value, and $\overline{y}_i$ is the mean of actual value.

Smaller values of rRMSE indicate that the forecasting models have better performance. The $R^2$ value ranges between 0 and 1. The closer the value of $R^2$ is to 1, the better the performance of the model [43].

## 4. Results and Analysis

The boll extraction results for all indices are presented in Section 4.1, and visual analysis was performed. The extraction effect of the index is quantitatively evaluated in Section 4.2. The correlation analysis of the extracted DCP with DTC and yield was performed, respectively, and the results show that our index is more correlated with yield, with a Pearson correlation coefficient of 0.84. The best index extraction results and the modeling effect of actual yield are shown in Section 4.3.

### 4.1. Extraction of Cotton Bolls

The cotton boll index combined with the Gaussian filter and Otsu method can be used to extract cotton bolls from the cotton field background. Thirty-four open cotton boll masks in the center area of the experimental field are shown in Figure 7.

By visual analysis, the extraction of the indices in C1 had the worst performance. There were many false negatives in b-r_d, b-r_r, and bg-r_r. The open cotton bolls in the original image were not masked. Many false positives appeared in g-r_r. Portions of the original image that were not open cotton bolls were also masked. The performance of bgr_sum was better than others, and the white cotton bolls in the original image were covered by the red mask.

The extraction based on the indices in C2 was significantly improved. The index extraction of all C2 combinations was relatively stable. The white boll pixels in the original image were covered by the red mask. As the number of bands participating in the construction of the index increased, the extraction performance improved.

The extraction performance of the indices in C3 was similar to that of the indices in C2. However, the performance of the difference method was poorer, and the false positives were obvious. The performance of other combinations was consistent with that of the extraction of indices in C2.

Compared with the difference and ratio methods, the index extraction performance of the normalized method was better. For example, in the combination of bgr-renir, there were many false negatives in the results of the difference method, the results of the ratio method were slightly better, and the results of the normalized method almost completely masked the boll pixels.

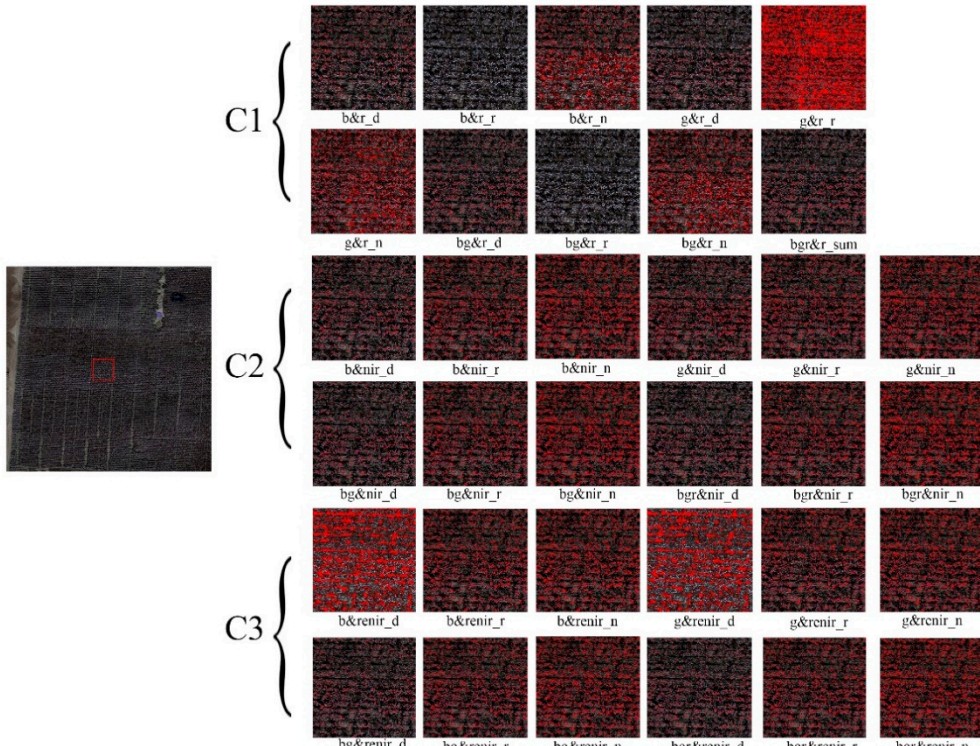

**Figure 7.** Thirty-four open cotton boll masks. Original images of a small area in the experimental field are shown. Thirty-four boll mask images generated by thirty-four indices were superimposed on the original image. C1, C2, and C3 indicate three band combinations.

Indices with multiple bands, such as bgr-nir_n, bgr-nir_r, and bgr-renir_n, performed better than indices with fewer bands, such as bg-nir_n, bg-nir_r, and bg-renir_n.

*4.2. Correlation Analysis*

The correlations between the results extracted by different indices and the field survey data were analyzed at the plot scale. Sections 4.2.1 and 4.2.2 show the correlation results for different indices with DTC and yield, respectively.

4.2.1. Correlation between DCP and DTC

The correlation analysis introduced in Section 3.2 was used to evaluate the performance of the indices in highlighting cotton bolls. The results are shown in Figure 8.

The b-r was not worth recommending. It exhibited the worst performance for all three calculation methods: difference, ratio, and normalization. The results showed that this band combination could not be used to highlight cotton bolls. However, several indices exhibited good performance, with correlation coefficients greater than 0.7, such as g-nir_n, bg-nir_n, bgr-nir_n, and bgr-renir_n.

All indices had some prominent effects on boll characteristics. The indices in C1 exhibited the worst performance, and the correlation coefficient was less than 0.4. The indices in C2 performed the best, with a correlation coefficient of 0.75. After the introduction of the red-edge band, the indices in C3 achieved performance similar to that of the indices in C2.

The four calculation methods exhibited a pattern in the indices constructed with the participation of infrared bands: the normalized calculation method had the highest correlation coefficient, the ratio method had the second-highest value, and the difference method had the lowest value.

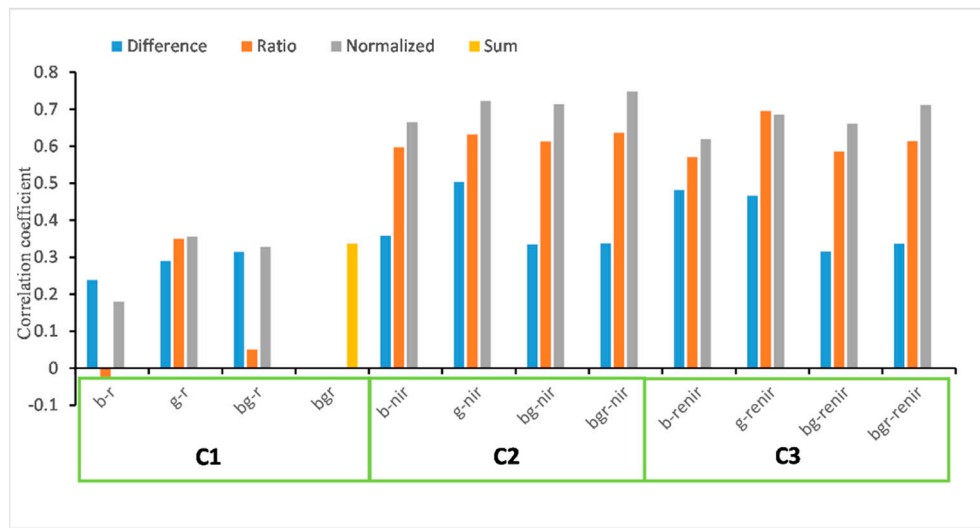

**Figure 8.** The correlation between DTC and DCP was extracted from each index. Three combinations and four calculation methods are included in the figure.

Bgr-nir_n achieved the best performance, and the correlation coefficient reached 0.75. It normalized the NIR band by the sum of blue, green, and red bands to highlight the characteristics of the high reflectivity of cotton bolls in the visible light band. Bgr-renir_n also achieved good performance, and the correlation coefficient reached 0.71. Vegetation exhibited wide variation in the red-edge band, so the inclusion of the red-edge band introduced uncertainty, which may explain why bgr-renir_n was inferior to bgr-nir_n.

4.2.2. Correlation between DCP and Yield

In this section, we analyze the relevance of the DCP extracted by each index and the yield based on the ground survey (Figure 9).

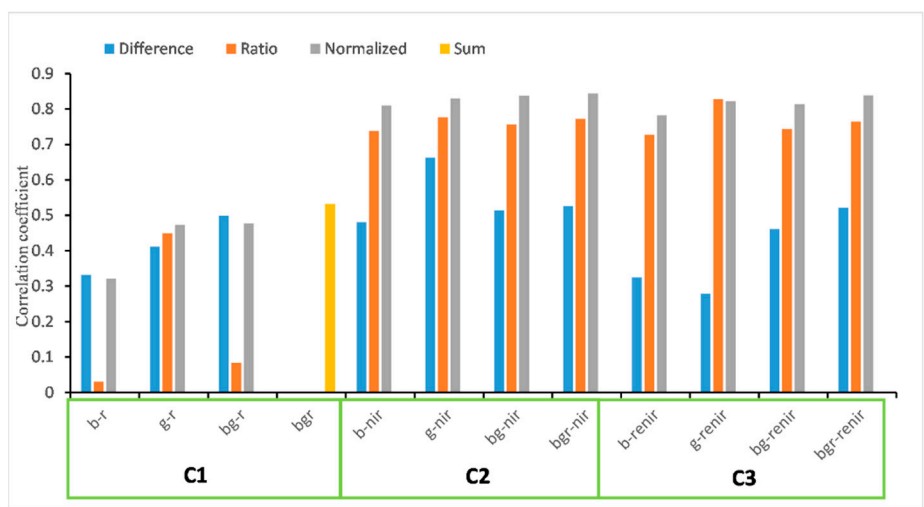

**Figure 9.** The correlation between yield and DCP was extracted from each index. Three combinations and four calculation methods are included in the figure.

All indices had a greater correlation with yield than DTC. Among them, the indices in C1 were the worst, and the correlation coefficients were all less than 0.53. However, once the infrared information was added, the correlation coefficient increased significantly. The correlation coefficient of the indices in C2 and C3 was as high as 0.84. The results for the indices in C2 and the indices in C3 were similar.

Among the three combinations, the index involving the red, green, and blue bands simultaneously obtained the best performance. In the C1 combination, bgr_sum had the best performance with a correlation coefficient of 0.53. In the C2 combination, bgr-nir_n had the highest correlation coefficient of 0.84. In the C3 combination, bgr-renir_n had the best performance, and the correlation coefficient was 0.84.

In this analysis, a pattern similar to that of the analysis of the correlation with DTC was observed; the correlation coefficient of the normalized calculation method was the highest, the ratio form resulted in the second-highest value, and the difference method had the lowest value.

### 4.3. The Results of Model Establishment and Accuracy Evaluation

Bgr-nir_n contributed to the construction and selection of the yield estimation model. Based on the results in Section 4.2, the correlation coefficient between bgr-nir_n and DTC was 0.75, which was the highest among all indices, and the correlation coefficient between bgr-nir_n and yield reached 0.84, which was also the highest. At the same time, considering the band acquisition capability of the sensor, the index using the near-infrared band and the visible light bands may have wider applications. Therefore, we recommend bgr-nir_n.

Five-fold cross-validation was used to determine the $R^2$ and rRMSE on the training set and testing set (Figure 10).

Our method provided an unbiased estimate of cotton yield. The predicted and actual values of the five models were distributed on both sides of the 1:1 line, and the distribution was relatively tight.

The results of the model assessment are summarized in Table 5.

**Table 5.** Assessment of the model fitting performance based on $R^2$ and rRMSE. For each column of data, the red–yellow–green level is used to indicate the quality of the evaluation metrics. Green is good, red is bad, and yellow is the middle.

| Model | Average | | | | Best | | | |
|---|---|---|---|---|---|---|---|---|
| | rRMSE Training | rRMSE Testing | $R^2$ Training | $R^2$ Testing | rRMSE Training | rRMSE Testing | $R^2$ Training | $R^2$ Testing |
| CART | 7.213 | 8.719 | 0.784 | 0.678 | 6.997 | 8.215 | 0.796 | 0.724 |
| KNN | 8.411 | 8.898 | 0.706 | 0.664 | 8.167 | 8.297 | 0.722 | 0.707 |
| RF | 7.478 | 8.367 | 0.767 | 0.704 | 7.250 | 7.560 | 0.781 | 0.755 |
| LR | 8.322 | 8.428 | 0.712 | 0.699 | 8.235 | 7.831 | 0.718 | 0.759 |
| SVR | 8.365 | 8.410 | 0.709 | 0.700 | 8.251 | 7.907 | 0.717 | 0.754 |

The best yield estimation was achieved by the random forest model. In five-fold cross-validation, the average $R^2$ reached 0.767 in the training set, and the average $R^2$ reached 0.704 in the testing set. The highest value of $R^2$ was 0.781 in the training set, and the highest value of $R^2$ was 0.755 in the test set. The average rRMSE was 7.478% and 8.367% in the training and testing sets, respectively.

The linear regression model and support vector regression model achieved similar performance. The average rRMSE reached 8.3% in the training set, and the average rRMSE reached 8.4% in the testing set. The average $R^2$ was approximately 0.71 in the training set, and the average $R^2$ exceeded 0.70 in the testing set.

There was obvious overfitting in the CART model. While the average $R^2$ reached 0.784 in the training set, it reached only 0.678 in the testing set. The average rRMSE reached 7.213% in the training set but reached only 8.719% in the testing set, which may demonstrate the high risk of overfitting by a single decision tree [44]. Random forest methodology solves the overfitting problem by increasing the number of trees.

The KNN model exhibited the worst performance. In five-fold cross-validation, the average $R^2$ only reached 0.706 in the training set, and the average $R^2$ reached 0.664 in the testing set. The average rRMSE was 8.411% and 8.898% in the training and testing sets, respectively.

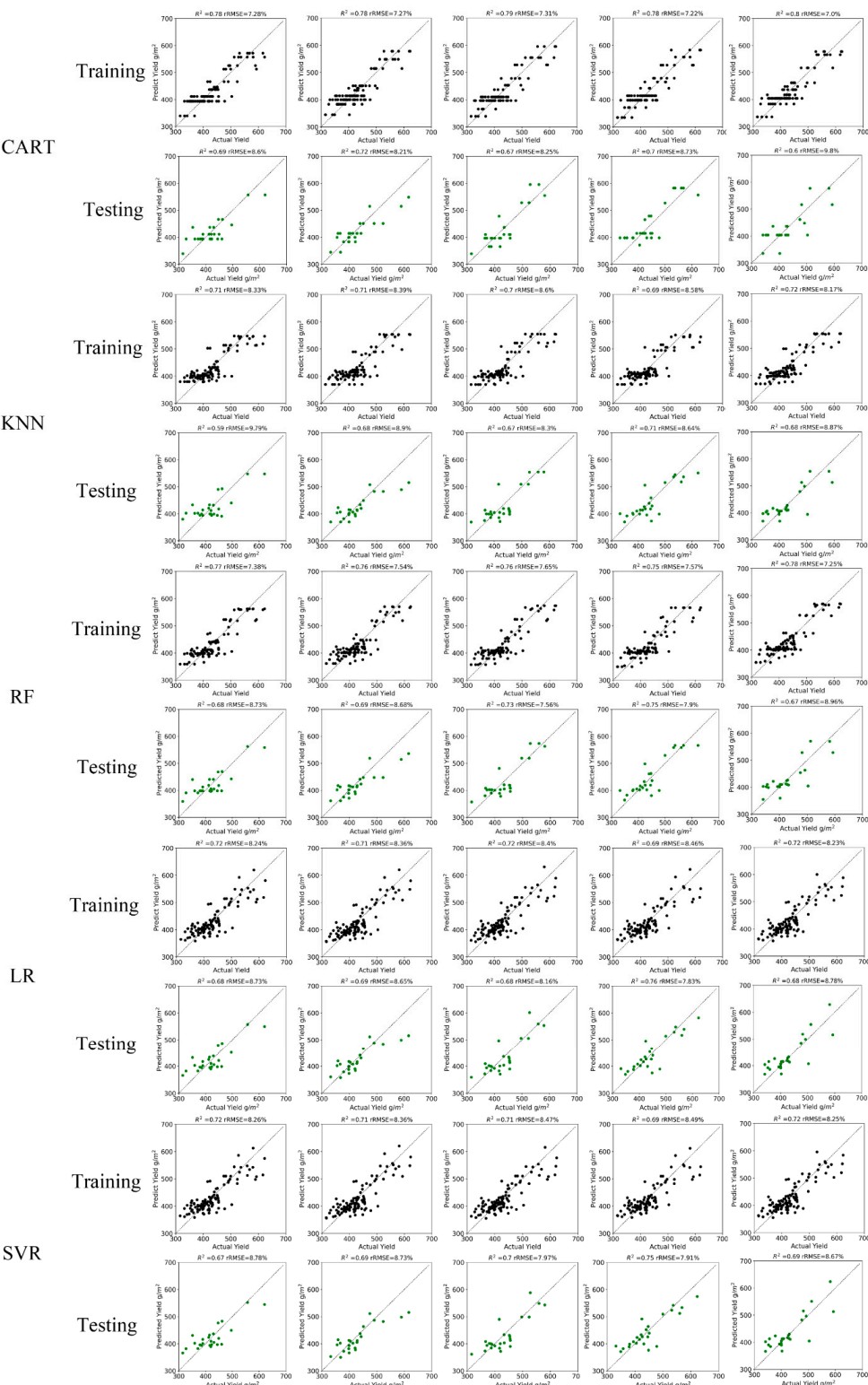

**Figure 10.** Five-fold cross-validation results of 5 models. Black dots represent training set results, and green dots represent testing set results.

The difference in performance in the training and testing sets was less apparent for the linear models. The average $R^2$ for both the linear regression and the SVR model was approximately 0.7 in both the training and testing sets. However, there was some difference in the average $R^2$ of the nonlinear model on the training set and the testing set, and the difference was less than 0.11.

The random forest method was applied to generate the yield map (Figure 11). The high-yield areas were concentrated in the northern part of the cotton field, and the southern part had lower yields. One possible reason is that better farmland management measures were implemented in the northern part of the cotton field.

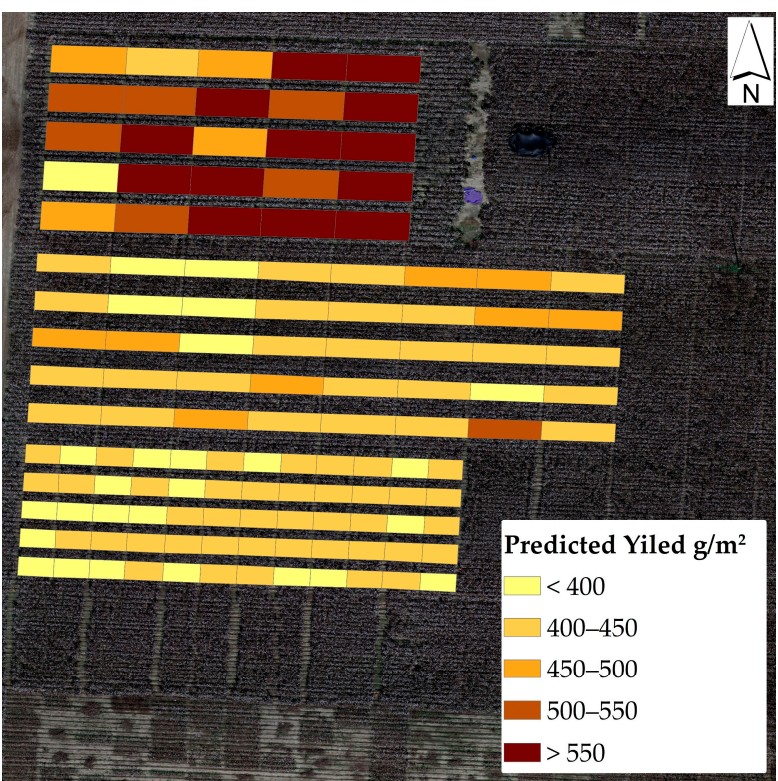

**Figure 11.** Yield map.

## 5. Discussion

The question of whether the remote sensing of cotton fields by UAVs will be blocked by the canopy was raised before the experiment. A conjecture was put forward that when cotton is close to harvest, cotton plants naturally senesce, the canopy shielding effect is weakened, and UAVs can monitor the entire plant, especially in cotton fields with defoliation and ripening treatments.

Our experiments in this paper confirmed this conjecture. In the later stage of cotton boll opening, our method extracted the cotton bolls off the whole plant, which contributed to the estimation of yield.

An additional experiment was designed to demonstrate the ability of UAVs to detect cotton bolls throughout the cotton plant rather than just the canopy. First, open upper cotton bolls were defined as the bolls harvested from the upper six branches of the cotton plant. The Number of Upper Cotton bolls (NUC) in all plots was obtained from the ground survey. The NUC was divided by the area of the experimental plot to obtain the Density of the Upper Cotton bolls (DUC). DUC represents the canopy information of cotton plants. DTC represents information for the entire cotton plant. The correlation between DCP and DUC was also analyzed (Figure 12). All index-extracted DCP were not relevant to DUC, and the correlation coefficients were less than 0.3. However, the experimental results in Section 4.2.1 show that the correlation coefficient between DCP and DTC reached 0.74. DCP was more relevant to DTC than DUC. This also proved that the entire plant is being monitored by the UAV and not just the canopy.

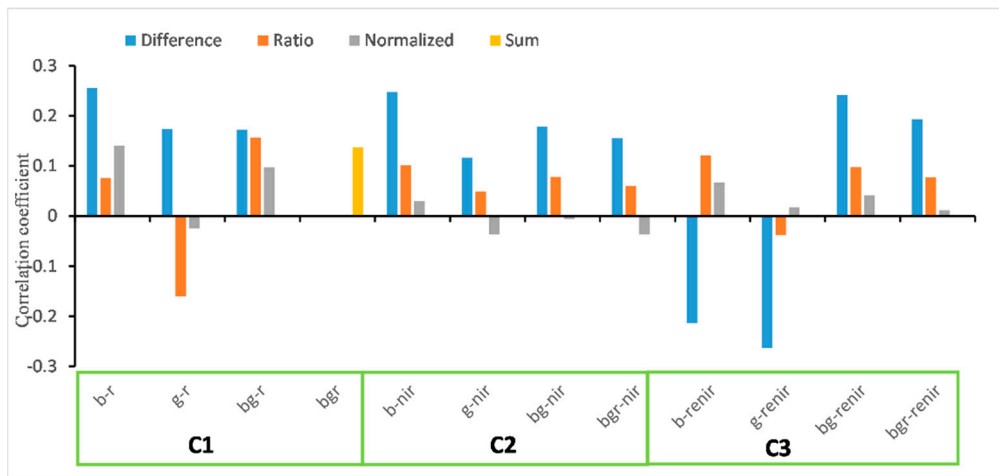

**Figure 12.** The correlation between DUC and DCP was extracted from each index. Three combinations and four calculation methods are included in the figure.

There was additional evidence (Figure 13) that UAV detection had some penetration capability in the preharvest period.

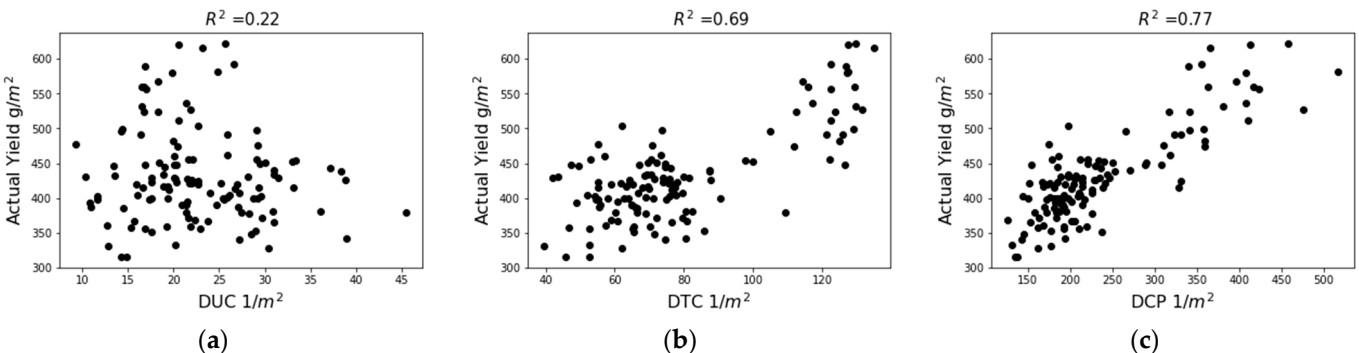

        (**a**)                         (**b**)                         (**c**)

**Figure 13.** (**a**) The relationship between DUC and actual yield. (**b**) The relationship between DTC and actual yield. (**c**) The relationship between DCP and actual yield. $R^2$ is achieved by RF. Each black dot represents an experimental plot.

The relationship between DUC and yield was weak. This is because the yield was harvested from all bolls and not just the upper bolls. The relationship between DTC and yield was slightly stronger. Our method is significantly closer to fitting yields with DTC than with DUC. This also showed that UAV remote sensing could provide information for the whole plant.

Since UAV remote sensing has a good ability to detect cotton bolls in cotton fields with defoliation and ripening treatment, there are many applications for UAVs to guide cotton production in the future, such as inversion of boll opening rate, evaluation of the defoliation effect, and yield prediction and estimation.

In the correlation analysis, we found that our method was more appropriate for yield than DTC. We considered boll size as a possible reason for this phenomenon.

In Figure 14, four pixels (black squares) are occupied by open cotton bolls of different sizes (red circles). All four of these pixels were identified as open cotton boll pixels, resulting in the same DCP corresponding to different DTC. However, the yield of the red circles shown in Figure 14a,b should be similar.

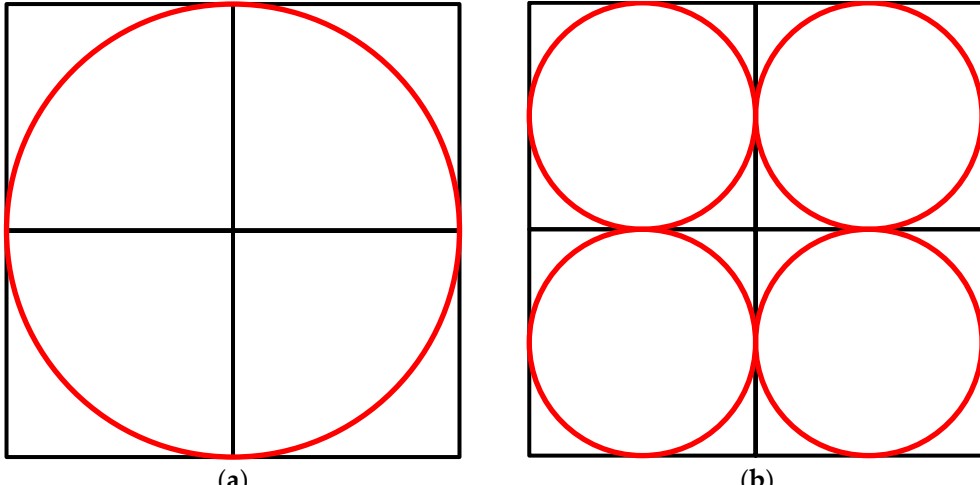

**Figure 14.** (**a**) Four pixels are occupied by one large boll. (**b**) Four pixels are occupied by four small bolls. They have similar yields, but very different boll counts.

This can also explain the phenomenon seen in Figure 14, where the relationship between DCP and yield is better than the relationship between DTC and yield. DCP represents the number of cotton boll pixels in the cotton field, and DTC represents the number of cotton bolls. DCP, to some extent, attenuates the effect of boll size on yield estimates. This suggests that our method may be more accurate than the traditional method of using boll count.

Our method is more accurate than the existing method that only uses the extraction of cotton bolls by threshold segmentation for yield estimation. OTSU and morphological filtering can extract boll area per grid, and this indicator was used to estimate cotton yield with $R^2$ up to 0.65 [16]. Laplacian thresholding was used to calculate the ratio of Cotton Unit Coverage (CUC), and this parameter was used for yield modeling. The $R^2$ reached 0.34 without removing outliers. The above two methods apply thresholding segmentation to the original reflectance [15]. In this paper, the DCP was obtained by calculating the index before threshold segmentation. Instead of using the original band directly, the cotton boll index highlighted the characteristics of the cotton bolls. Spectral information was fully utilized so that the average $R^2$ of five cross-validations obtained by our method was 0.77.

## 6. Conclusions

Previous studies on the extraction of cotton bolls directly used the reflectivity of the original band. A more direct approach is to design an index based on spectral characteristics to extract cotton bolls. In this paper, remote sensing indices were calculated by multispectral UAV images to highlight the spectral characteristics of cotton bolls, and the open cotton bolls were extracted by threshold segmentation. The performance of different indices was compared using correlation analysis. The index with the best performance and the yield from the field survey was modeled. The main conclusions of our study are as follows:

1.  Bgr-nir_n, an index that normalizes the NIR band by the sum of blue, green, and red bands, exhibited the best performance in highlighting information on open cotton bolls. The correlation coefficient between DCP extracted from this index and the measured yield was 0.84;
2.  DCP extracted from bgr-nir_n combined with the random forest methodology achieved an unbiased estimation of cotton yield on the plot scale; the average $R^2$ based on five-fold cross-validation was 0.77, and the average rRMSE was 7.5%.

The results of this study support a new approach to extracting cotton bolls by calculating the remote sensing index. Our method exhibited higher estimation accuracy than the existing methods that only use the extraction of cotton bolls for yield estimation. However, our method is only applied in the area where the experimental field is located. If satisfactory accuracy can be obtained in other areas, it will show that our method is robust to regional changes.

**Author Contributions:** Conceptualization, G.S.; Data curation, G.S.; Formal analysis, G.S.; Funding acquisition, Q.L.; Investigation, G.S., X.D. and Y.R.; Methodology, G.S. and X.D.; Project administration, X.D. and M.D.; Resources, H.W.; Software, G.S.; Supervision, Q.L.; Validation, G.S., Q.L. and X.T.; Visualization, G.S.; Writing—original draft, G.S. and Y.Z.; Writing—review and editing, G.S. All authors have read and agreed to the published version of the manuscript.

**Funding:** This research was funded by the National Science Foundation of China (42071403) and the Key Program of High-resolution Earth Observation System (00-Y30B01-9001-22/23, 20-Y30F10-9001-20/22).

**Institutional Review Board Statement:** Not applicable.

**Informed Consent Statement:** Not applicable.

**Data Availability Statement:** Not applicable.

**Conflicts of Interest:** The authors declare no conflict of interest.

## Appendix A

**Table A1.** Field A survey data.

| ID | NTC | Actual Production/g |
|---|---|---|
| A1 | 5663.87 | 22,140.00 |
| A2 | 5493.60 | 20,500.00 |
| A3 | 5057.60 | 23,550.00 |
| A4 | 5819.67 | 25,170.00 |
| A5 | 5140.80 | 25,570.00 |
| A6 | 5031.00 | 21,360.00 |
| A7 | 5627.53 | 21,680.00 |
| A8 | 5925.33 | 23,750.00 |
| A9 | 5704.67 | 26,530.00 |
| A10 | 5510.27 | 25,060.00 |
| A11 | 5804.93 | 22,480.00 |
| A12 | 5279.60 | 24,160.00 |
| A13 | 4728.00 | 22,330.00 |
| A14 | 5738.53 | 26,160.00 |
| A15 | 5720.40 | 26,110.00 |
| A16 | 5691.73 | 20,180.00 |
| A17 | 5516.00 | 23,050.00 |
| A18 | 5211.60 | 25,210.00 |
| A19 | 5514.13 | 26,690.00 |
| A20 | 5831.20 | 28,030.00 |
| A21 | 5456.07 | 22,080.00 |
| A22 | 5566.25 | 23,570.00 |
| A23 | 5842.60 | 23,920.00 |
| A24 | 5740.00 | 27,900.00 |
| A25 | 6072.00 | 27,750.00 |

**Table A2.** Field B survey data.

| ID | NTC | Actual Production/g |
|----|-----|---------------------|
| B1 | 1645.00 | 10,381.76 |
| B2 | 2094.20 | 9973.11 |
| B3 | 1953.00 | 9536.92 |
| B4 | 1911.60 | 11,298.52 |
| B5 | 2739.10 | 12,377.12 |
| B6 | 1938.00 | 13,010.75 |
| B7 | 1968.00 | 12,341.12 |
| B8 | 2212.50 | 10,028.61 |
| B9 | 1887.60 | 10,331.08 |
| B10 | 2352.00 | 9645.34 |
| B11 | 2042.40 | 9332.29 |
| B12 | 2074.60 | 11,015.15 |
| B13 | 1829.00 | 11,137.88 |
| B14 | 1903.50 | 12,335.85 |
| B15 | 1834.00 | 12,233.03 |
| B16 | 2002.00 | 12,635.14 |
| B17 | 1864.20 | 11,373.58 |
| B18 | 2065.00 | 11,612.64 |
| B19 | 2989.60 | 10,365.61 |
| B20 | 2038.20 | 12,290.72 |
| B21 | 2673.20 | 12,416.68 |
| B22 | 1753.80 | 11,543.43 |
| B23 | 1512.80 | 10,899.73 |
| B24 | 2204.60 | 10,413.39 |
| B25 | 2201.90 | 9340.95 |
| B26 | 2256.00 | 10,398.35 |
| B27 | 2401.60 | 11,656.75 |
| B28 | 2390.10 | 12,038.14 |
| B29 | 2394.40 | 12,011.73 |
| B30 | 2081.20 | 12,116.70 |
| B31 | 1686.40 | 11,516.07 |
| B32 | 2186.80 | 11,802.84 |
| B33 | 1800.00 | 10,589.28 |
| B34 | 1152.40 | 11,754.78 |
| B35 | 1292.00 | 12,231.23 |
| B36 | 2232.10 | 11,724.94 |
| B37 | 1898.00 | 11,873.73 |
| B38 | 1928.00 | 12,451.56 |
| B39 | 2016.90 | 13,596.48 |
| B40 | 2175.40 | 11,160.55 |

**Table A3.** Field C survey data.

| ID | NTC | Actual Production/g |
|---|---|---|
| C1 | 928.20 | 5110.38 |
| C2 | 967.60 | 4517.57 |
| C3 | 854.90 | 4372.02 |
| C4 | 809.10 | 4613.21 |
| C5 | 658.00 | 4892.43 |
| C6 | 834.20 | 5264.71 |
| C7 | 674.50 | 4704.62 |
| C8 | 782.00 | 5184.57 |
| C9 | 868.00 | 5201.47 |
| C10 | 529.30 | 5246.20 |
| C11 | 669.90 | 5809.46 |
| C12 | 671.50 | 5138.40 |
| C13 | 676.80 | 4737.80 |
| C14 | 638.40 | 4323.09 |
| C15 | 569.50 | 4344.34 |
| C16 | 790.50 | 4739.49 |
| C17 | 696.60 | 4376.91 |
| C18 | 930.60 | 5008.68 |
| C19 | 907.80 | 5400.04 |
| C20 | 693.60 | 5593.89 |
| C21 | 761.40 | 4861.65 |
| C22 | 754.80 | 6119.89 |
| C23 | 940.50 | 5043.02 |
| C24 | 643.20 | 5541.37 |
| C25 | 756.00 | 3979.26 |
| C26 | 639.60 | 4034.90 |
| C27 | 481.00 | 4030.50 |
| C28 | 768.00 | 5069.61 |
| C29 | 739.50 | 4459.73 |
| C30 | 846.00 | 5057.83 |
| C31 | 717.80 | 5099.76 |
| C32 | 943.40 | 4919.87 |
| C33 | 1100.00 | 4863.21 |
| C34 | 921.50 | 5200.45 |
| C35 | 897.60 | 5133.75 |
| C36 | 789.60 | 4736.87 |
| C37 | 916.70 | 5116.49 |
| C38 | 795.60 | 4361.68 |
| C39 | 791.70 | 4330.99 |
| C40 | 875.50 | 4513.94 |
| C41 | 797.90 | 4274.79 |
| C42 | 897.60 | 5219.46 |
| C43 | 911.80 | 4838.37 |
| C44 | 949.40 | 5150.25 |
| C45 | 865.20 | 4860.50 |
| C46 | 738.00 | 4792.67 |
| C47 | 632.40 | 4907.05 |
| C48 | 777.60 | 4858.27 |
| C49 | 717.80 | 4474.98 |
| C50 | 556.80 | 3831.51 |
| C51 | 639.00 | 3843.09 |
| C52 | 671.60 | 5055.71 |
| C53 | 703.10 | 4854.22 |
| C54 | 809.60 | 4684.16 |
| C55 | 595.90 | 4780.87 |
| C56 | 791.20 | 5104.70 |
| C57 | 838.30 | 4868.06 |
| C58 | 855.60 | 5268.38 |
| C59 | 600.60 | 5418.95 |
| C60 | 662.40 | 4837.42 |

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
