# Peer review of "Cotton Yield Estimation Using the Remotely Sensed Cotton Boll Index from UAV Images"

_drones, doi:10.3390/drones6090254_

Round 1

Reviewer 1 Report

The manuscript by Shi et al. presents results from a study estimating cotton yield using the remotely sensed cotton boll indices from UAV images. The manuscript is well written and is easy to follow. No concerns were identified during the review. The results are clear and deserve to be published.

Author Response

Response to reviewer1 comments is attachment, but in case of not can open it, there is:

Response to reviewer1 comments

Comments: The manuscript by Shi et al. presents results from a study estimating cotton yield using the remotely sensed cotton boll indices from UAV images. The manuscript is well written and is easy to follow. No concerns were identified during the review. The results are clear and deserve to be published.

Response: Thank you for your dedication to the review. Your comments encourage us a lot. Thanks again for your valuable comments on our manuscript.

Reviewer 2 Report

The review presents a paper aimed at developing a method for estimating cotton yield based on multispectral survey data. The authors have done a tremendous job of developing an index estimating the number of cotton balls. The work is done at a high level and deserves to be published in the Drones journal.

Small remarks:

1) the Abstract does not clearly indicate the purpose of the study, it should be added.

2) The Introduction refers only to 11 literary sources. There is much more work aimed at studying the yield of cotton according to UAV and satellite images. It is advisable to supplement the review.

3) Lines 89-90 Represent the conclusion about the work done, it is possible to rephrase or move to another section

4) Line 271, 286, 310. Add abbreviations or full transcription for CART and KNN. Support vector regression - add an abbreviation on line 271 and remove it from line 286. Line 310 - add an abbreviation on line 271 and remove it from line 310.

5) It is necessary to revise the design of literary sources and, if possible, supplement them.

Author Response

Dear Reviewer,

Thank you very much for your time involved in reviewing the manuscript and your very encouraging comments on the merits. We have revised the manuscript according to your suggestions, The response to you is in the attachment.

Thanks again for your comments.

Yours sincerely,
Guanwei Shi

Reviewer 3 Report

The Comments and Suggestions are attachment, but in case of not can open it, there is:

This work describes a system capable of estimating the amount of mature cotton present in a study area. The system uses a UAV robot with a spectral image sensor to determine mature cotton, so a support decision for the harvest is expected based on the system's response. The work demonstrates severe shortcomings in the methodology description, mainly in the scientific aspect of analyzing the amount of cotton produced with the one identified by the UAV images. Despite this, the algorithms used for image classification resulted in high accuracy and satisfactory descriptions of their methods. However, the algorithms do not have innovations in this application, that is, algorithms are used without any analysis of their best parameters.

The abstract and keywords can be better explained so that future readers can understand the work in a first interaction, suggestions:

·         The keywords used are generic (cotton, UAV, Yield estimation). In this sense, the authors should be more specific for the work to be easily found by search engines in the future.

·         In the Abstract, there are two questions: why are some acronyms described (DCP and DTC) and others not (SVR, CART, RF, and KNN)? And what kind of correlation is used at work? (Line 20: "correlation coefficient of 0.84".).

·         Lines 22 to 24 are confusing. The sequence of ideas described makes the reader get lost in the reading. Try changing the meaning of the sentence.

·         I still suggest not using a proper term in the Abstract (e.g., bgr&nir_normalized), as the reader will not yet know the meaning of this. Instead, I suggest using: "RGB and NIR normalized".

The work has some formatting errors that can be found at:

·         Line 6 has a "; " the first in affiliation.

·         All acronyms should be revised with capital letters; for example, "the density of open cotton boll pixels (DCP)" should be declared as "the Density of open Cotton boll Pixels (DCP)".

·         Line 38: unmanned aerial vehicles (UAVs) > Unmanned Aerial Vehicles (UAVs).

·         Line 40: what is EVI? And where is the citation of this work?

·         Line 42: what are SPOT 5, TM, and ETM?

·         Line 50: the acquisition of data > data acquisition

·         Line 75: double space

·         Line 87: Our > our

·         Line 116: red-edge(RE) > Red-Edge (RE)

·         Line 116: near infrared red(NIR) > Near Infrared Red (NIR)

·         Line 119: ground sampling distance (GSD) > Ground Sampling Distance (GSD)

·         Line 128: Captions should not be separated from tables.

·         Line 130: the number of total open cotton bolls (NTC) > the Number of Total open Cotton bolls (NTC)

·         Line 227: The formula for a 2D Gaussian filter is[18] > The Equation for a 2D Gaussian filter is [18]

·         Line 250: DCP has been described before, no need to write in full again

·         Line 281: What is Eq.(18)? Could it be Equation 18? If yes, where is this equation?

·         Line 512: Where is figure 14?

·         Lack of spaces between the letters and citations' brackets (whole work).

As for the organization of the text, the choice of names and sequence of sections is adequate. However, the Introduction is weak, addresses the topic with short elaborations on new ideas based on the indicated references. There is no section dedicated to state of the art on the topic addressed. I also recommend that authors consider writing introductory texts at the beginning of each section and subsection, for example: between lines 94 and 95, between lines 110 and 111, between lines 164 and 165, between lines 336 and 337, between lines 367 and 368.

Some statements are made without being based on a bibliographic reference, and some parts need to be detailed, such as:

·           A problem in the statement between lines 28 and 33: what is FAO? Use a citation to affirm the data described in these lines.

·           A problem in the study description, line 101, the total of fields is 125, and there can be three types. Therefore, I would suggest explaining each type of field's quantities and naming them distinctly.

·           Use a citation to affirm the data described between lines 105 and 109.

·           Table 2 does not provide relevant information and is too generic for the studied area's size.

·           Line 134 states that all fields were harvested and weighed, so why not make a table with the identified areas (differentiating each of the three field types) and the amounts of cotton harvested? For example, put a letter for each field and list cotton production in a table. Field A produced 1 kilogram of cotton, Field B had 2 kilograms, and so on. If the table for 125 fields gets long, why not split it into three tables?

·           Improve the quality of Figure 3

·           Line 173: the figure caption says that 65,603 cotton boll pixels and 118,996 were analyzed. How is it possible to analyze half of a cotton boll? Or half a pixel? Are there any numerical representation errors?

·           Line 242: This correlation should be stated in the Abstract.

·           The original image of Figure 5 could be demonstrated before Table 3, so the reader could analyze the data in Table 3 side by side with the spectral image.

·           Assertion problem on line 311: "Ensemble learning is a new approach to machine learning". Is it recent? This technique has been used for almost 20 years.

At the end of the reading, there were still four questions to be clarified:

·         Shouldn't Figure 13 be the explanation of the method? If so, I would suggest moving Figure 13 to Section 3, and the description of the algorithm recognizes cotton bolls.

·         If the best option pointed out is to use Random Forest, why in Figure 12 contain an R2 generated with linear regression?

·         RGB analysis not done before and after cotton harvest? Wouldn't that be the easiest way to do subsection 3.1.1? Was there only one flight with the UAV? Otherwise, the authors should explain the reason for having performed only one test of the spectral analysis.

·         What kind of robustness do authors intend to improve? (line 550)

Author Response

Dear Reviewer:

Thank you very much for your time involved in reviewing the manuscript and your very comments. We have revised the manuscript according to your suggestions, The response to you is in the attachment.

Thanks again for your comments.  

Yours sincerely,

Guanwei Shi

Round 2

Reviewer 3 Report

Dear Authors,

Thank you for being very professional in each topic of my comments and suggestions. I read version 2, and its sounds better! It is very uncommon to see authors improving/accepting all suggestions. Congratulations!

Best regards.